# Freeze–Thaw Pre-Treatment of Cassava Tubers to Improve Efficiency of Mechanical Peeling

**Ziba Barati** *, **Sajid Latif**, **Sebastian Romuli** and **Joachim Müller**

Institute of Agriculture Engineering (440e), Tropics and Subtropics Group, University of Hohenheim, 70599 Stuttgart, Germany

* Correspondence: Barati@uni-hohenheim.de; Tel.: + 49-711-459-24704; Fax: + 49-711-459-23298

**Abstract:** The effect of a freeze–thaw pre-treatment (FTP) on the peeling process of cassava tubers was investigated in this study. The length and weight of the cassava tubers varied from 200 to 280 mm and 500 to 900 g, respectively. A prototype abrasive cassava peeling machine was used. The operational parameters were the rotational speed of the brushes (550–1150 rpm), peeling time (1–5 min), thawing temperature (50–90 °C), and incubation time of the thawing treatment (0–120 s). Response surface methodology was applied to optimize FTP to improve the peeling process of cassava tubers. Peeled surface area and peel loss were measured as the responses. Results revealed that the peeled surface area and peel loss were significantly influenced by the rotational speed of the brushes, peeling time, and the incubation time of the thawing treatment ($p < 0.05$). Under optimal peeling conditions, with a rotational speed of 1000 rpm, a peeling time of 3.4 min, a thawing temperature of 59 °C, and an incubation time of 90 s, the peeled surface and the peel loss were approximately 99.5 and 19%, respectively. The results show that the use of FTP can improve cassava peeling by softening the peels and increasing the peeled surface area.

**Keywords:** freeze–thaw pre-treatment; mechanical peeling; peel loss; peeling efficiency; response surface method

## 1. Introduction

Cassava, one of the staple foods in tropical and sub-tropical lands, has gained attention due to its capability to provide food security in recent decades [1–3]. Fresh cassava tubers do not have a long shelf-life due to their high moisture content [4]. Therefore, cassava tubers are usually processed to obtain various relatively shelf-stable products [5]. Peeling is an essential step in cassava processing, which includes removing the corky periderm and cortex from the tubers.

Peeling of cassava tubers, either for industrial or domestic use, is still a major challenge in cassava processing. Currently, it is traditionally done by hand or by mechanical, thermal, or chemical methods [6]. Practically, each method of peeling presents its own advantages and disadvantages. Traditional manual peeling is slow and labor intensive. Mechanical peeling of cassava tubers done in small and large-scale industries creates high losses and has an inefficient peeling rate. The peeling efficiency decreased in mechanical peeling [6] by increasing the rotational speed of the peeling tools (brushes or drums) and the peeling time. The technical data on the properties of cassava tubers to design an appropriate peeling machine is insufficient. Furthermore, irregular shape, size, age, and different varieties of cassava tubers affect the peeling process [7–9].

Despite several attempts to develop a mechanized peeling machine in Nigeria, Brazil, and China, no efficient cassava peeling machine is currently available on the market [9–12]. This can be explained by the broad variations in size, weight, and peel thickness of cassava tubers as well as the irregularity in their shape [7,13,14]. Moreover, environmental factors such as relative humidity, temperature, rainfall,

type of soil, moisture content of soil, acidity of soil, fertility of soil, and vegetation on the farm may affect the characteristics of cassava tubers that impact the peeling process [15]. Considering all the problems, Ohwovoriole et al. [10] stated that an effective cassava peeling machine should remove the cortex of the tuber efficiently without considerable loss of flesh.

Previous studies have shown that the freeze–thaw method could improve the peeling process as achieved in tomato processing [16,17]. It was found that 20 to 30 s of chilling process in cooled brine continued by further thawing in hot water could loosen the peels for easy removal [18].

However, to the best of our knowledge so far no studies were published on the application of the freeze–thaw pre-treatment for the peeling process of cassava tubers. Accordingly, the main objectives of this study were: (1) to investigate the impact of freeze–thawing on the peeling of cassava tubers, (2) to optimize the freeze–thaw pre-treatment to improve the peeling process, and (3) to investigate the effect of freeze–thaw pre-treatment on the quality of peeled cassava tubers.

## 2. Materials and Methods

### 2.1. Material

Cassava tubers imported from Costa Rica were purchased from the local market in Stuttgart. The tubers were chosen based on their length, mass, and diameter. The selection criteria were tuber lengths of 200 to 280 mm, a tuber mass of 500 to 900 g, and tuber diameters of 60 to 80 mm. After screening, the cassava tubers were stored at $-18$ °C for the experiment.

### 2.2. Characterization of Cassava Tubers

Before sorting the cassava tubers for the peeling experiment, they were characterized by their peel thickness, dry matter of the peels, dry matter of the flesh of tuber, proportion of peel mass of the tuber, force to penetrate the peel, and force to penetrate the tuber flesh. The thickness of the peels was measured with an accuracy of 0.1 mm by using a Vernier caliper. The penetration force was measured by a penetrometer (PCE-FM200, PCE Deutschland GmbH, Meschede; Germany). Dry matter content was determined based on DIN CEN/TS 14774-3 [19] using a cabinet dryer (UM 700Memmert GmbH & Co.KG, Schwabach, Germany). The peel mass proportion of the tuber was calculated as peel loss *PL*:

$$PL = \frac{m_1 - m_2}{m_1} \cdot 100 \tag{1}$$

where *PL* is the proportion by weight of the tuber peel (%), $m_1$ is the mass of the tuber before peeling (g), and $m_2$ is the mass of the peeled tuber (g).

As for control value, the mass of peel when the tuber is carefully manually peeled to avoid flesh loss $PL_{man}$ was also measured.

The characteristic parameters were measured for 20 cassava tubers and the mean values were presented.

### 2.3. Experimental Procedure

#### 2.3.1. Description of the Prototype Cassava Peeling Machine

A prototype abrasive cassava peeling machine was used. The peeling machine was 1500 mm long, 500 mm wide, and had a height of 1000 mm. The machine consisted of five rotating cylinders, which were covered by abrasive brushes (ZZB10022-439648, August Mink KG, Göppingen, Germany) (Figure 1). Other components of the machine were frame, water bath with adjustable heating system (UNOLD 58815, Conrad, Hirschau, Germany), motor (1.5 kW, 1LA 5090, Siemens AG, München, Germany), and frequency converter (ST 8100, Sourcetronic GmbH, Bremen, Germany) to regulate the rotational speed of the brushes.

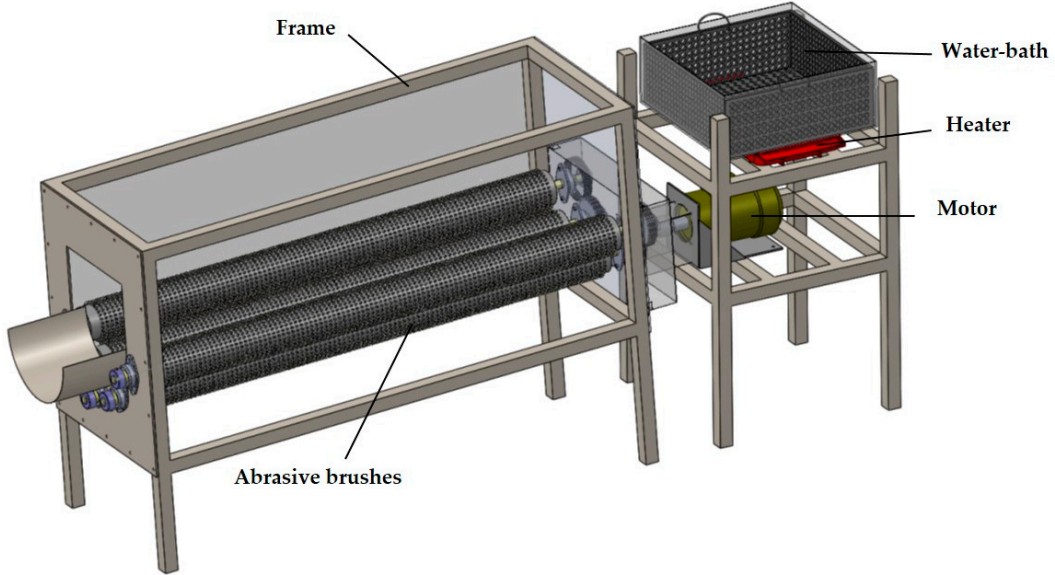

**Figure 1.** The prototype cassava peeling machine.

### 2.3.2. The Freeze–Thaw Pre-Treatment (FTP)

Before peeling with the abrasive peeling machine, cassava tubers were frozen at −18 °C for 24 h and treated in a water-bath by applying different temperatures (50, 60, 70, 80, and 90 °C) and incubation times (0, 30, 60, 90, and 120 s).

### 2.3.3. Peeling Process

After the freeze–thaw pre-treatment (FTP), the treated cassava tubers were peeled using the prototype abrasive peeling machine by applying different rotational speeds (550, 700, 850, 1000, and 1150 rpm) and peeling times (1, 2, 3, 4, and 5 min). For each parameter combination, 3 cassava tubers were treated and peeled using the peeling machine, and the mean values of these 3 tubers are reported in the study.

### 2.4. Experimental Design

Response surface methodology (RSM) using central composite design (CCD) was elaborated to study the effect of FTP under variation of rotational speed of the brushes, peeling time, thawing temperature, and incubation time on the peeling process for cassava tubers.

The CCD consisting of 30 combinations (6 central points), with a four-level full factorial design using coded factor −2, −1, 0, +1, and +2 was applied in this study. The selected independent variables and limit levels for the response surface study are presented in Table 1. The peeled surface area and peel loss were chosen as response factors for evaluating the cassava tuber peeling process. It was assumed that maximizing the peeled surface area to 100% and peel loss ≤20%, where 15 to 20% represents the typical proportion of peels in the tuber [11,20], would result in an optimum peeling process.

The peeled surface area was determined according to Srikaeo, Khamphu, and Weerakul [21] by analyzing the photos, which were taken from cassava tubers after the peeling process, using image processing software (Fiji, Madison, WI, USA). The peeled surface area (PSA) was calculated as:

$$PSA = \frac{A_1}{A_2} \cdot 100 \qquad (2)$$

where $PSA$ is the peeled surface area after the peeling process (%), $A_1$ is the area of the removed peel on the tuber surface (cm$^2$), and $A_2$ is the whole cassava tuber surface (cm$^2$).

The peel loss was computed according to Equation (1).

**Table 1.** Independent variables and limit levels for response surface study.

| Variables | Unit | Coded Factors | Levels | | | | |
|---|---|---|---|---|---|---|---|
| | | | −2 | −1 | 0 | 1 | 2 |
| Rotational speed of brushes | rpm | $X_1$ | 550 | 700 | 850 | 1000 | 1150 |
| Peeling time | min | $X_2$ | 1 | 2 | 3 | 4 | 5 |
| Thawing temperature | °C | $X_3$ | 50 | 60 | 70 | 80 | 90 |
| Incubation time | s | $X_4$ | 0 | 30 | 60 | 90 | 120 |

## 2.5. Starch Content Analysis

To investigate the characteristics of the peeled cassava tubers, the total starch content of manually peeled cassava tubers and peeled cassava tubers after FTP using the peeling machine was determined by R-Biopharm assay kit (Nr. 10 207 748 035, R-Biopharm AG, 64297 Darmstadt, Germany) following AOAC method No. 996.11 [22].

## 2.6. Scanning Electron Microscopy of Freeze–Thaw Treated Cassava Tubers

To investigate the effect of FTP on the structure of cassava starch, scanning electron microscopy (SEM) images were obtained from the manually peeled cassava tubers as a control of this study and from the peeled cassava tubers after FTP using the peeling machine at optimum conditions. The SEM was conducted according to Ayetigbo et al. [23]. The vacuum dried samples at 40 °C were grounded and directly placed on a graphite layer on a gold-plated cylinder. The samples were observed at magnifications of ×200–2000 in a scanning electron microscope (JSM-IT100, JEOL GmbH, Freising, Germany) under high-vacuum conditions with an accelerating voltage of 2.5 to 20 kV. To confirm the reproducibility of the results, at least 5 images were obtained from different areas of the samples.

## 2.7. Statistical Analysis

A student version of Design Expert 11 (STATCON GmbH, Witzenhausen, Germany) was used to design the experiments and to analyze the obtained data. The data was originally analyzed with the full cubic model for each response. The full models were later adjusted by removing the statistically insignificant terms, not considering those required to support hierarchy. The final reduced cubic models were then determined. The analysis of variance (ANOVA) was applied to examine the significance of independent variables and their interactions at $p$ value < 0.05 (95% confidence level). The accuracy of the mathematical model was estimated using statistical analysis of coefficient of correlation ($R^2$) and mean absolute percentage error (*MAPE*). In order to validate the optimization of cassava tuber peeling by FTP through the peeling machine, three replicates were conducted under optimum conditions.

## 3. Results and Discussion

### 3.1. Characteristics of Cassava Tubers

The characterization of cassava tubers in this study is presented in Table 2. The mass proportion of the peels in the cassava tubers ranged from 13.3 to 15.6%, which is in accordance to the results (8.5 to 17%) reported by Ezekwe [13]. The observed thickness of the peels ranged from 2.7 to 3.3 mm in this study. A wider thickness range for peels of 1.2 to 4.1 mm was reported by Adetan et al. [11]. The range of diameter and length of cassava tubers varied from 53.1 to 90.9 mm and from 194 to 320 mm, respectively. The force necessary to penetrate the cassava tuber peels and cassava tuber flesh varied from 4.4 to 5.6 (N/mm$^2$) and from 2.8 to 3.8 (N/mm$^2$), respectively. Similarly, a peel penetration force of 3.3 to 5.47 (N/mm$^2$) for cassava tubers was reported by Adetan et al. [11]. Other characteristic parameters including the weight of the tuber, the dry matter of the peel, and the dry matter of tuber in this study are similar to previous studies with slight differences [11,24].

**Table 2.** Characteristics of cassava tubers (n = 20).

| Parameters | Mean ± SD |
| --- | --- |
| Mass of tuber (g) | 733.4 ± 254.7 |
| Length of tuber (mm) | 240 ± 40.0 |
| Diameter of tuber (mm) | 66.7 ± 9.9 |
| Peel thickness (mm) | 2.9 ± 0.2 |
| Penetration force, flesh (N/mm$^2$) | 3.3 ± 0.7 |
| Penetration force, peel (N/mm$^2$) | 5.0 ± 0.4 |
| Dry matter, flesh (%) | 32.9 ± 4.1 |
| Dry matter, peel (%) | 43.3 ± 19.4 |
| Proportion of peel by mass (%) | 14.7 ± 0.9 |

A weight range of about 900 g (415–1287 g) was observed in the sample, associated with the mean ± SD of 733.4 ± 254.7. Sorting of the agricultural products prior to packaging or processing is a routine in post-harvest operations. Therefore, based on the properties of the normal distribution and after checking for normality, the weight range of the tubers was limited to 500–900 g, which as approximation of "mean ± 1 × SD" should cover more than two- thirds of the population.

*3.2. Effect of the Rotational Speed of the Brushes, the Peeling Time, and FTP on the Peeling Process for Cassava Tubers*

The RSM design matrix for the rotational speed of the brushes, the peeling time, and FTP are presented in Table 3 together with the peeled surface area and the peel loss. The peeled surface area and peel loss ranged from 14.9 to 97.9% and 4.2 to 37.5%, respectively, by variation of condition parameters (rotational speed of brushes, peeling time, thawing temperature, and incubation time). The average peeled surface area and peel loss of cassava tubers was 64.1 and 18.7%, respectively.

The mathematical equation obtained from RSM for the peeled surface area (PSA) of cassava tubers is as follows:

$$
\begin{aligned}
PSA = -148.87408 &+ 0.135258{\cdot}v - 14.11093{\cdot}t_p + 2.4693{\cdot}T - 3.72594{\cdot}t_t + 0.135243 \\
&{\cdot}v{\cdot}t_p + 0.005847{\cdot}v{\cdot}t_t - 0.76666{\cdot}t_p{\cdot}T + 1.61357{\cdot}t_p{\cdot}t_t - 0.000283{\cdot}v^2 \\
&- 6.01193{\cdot}t_p^2 - 0.006776{\cdot}t_t^2 - 0.001856{\cdot}v{\cdot}t_p{\cdot}t_t
\end{aligned} \tag{3}
$$

where *PSA* is the peeled surface area after FTP and the peeling process (%), $v$ is the rotational speed of the brushes (rpm), $t_p$ is the peeling time (min), $T$ is the thawing temperature (°C), and $t_t$ is the incubation time (s).

It was observed that increasing the rotational speed of the brushes, the peeling time, thawing temperature, and incubation time had a positive effect on the peeled surface area.

The effects of individual variables and their interaction on the peeled surface area are shown in Table 4. The accuracy of the model was indicated by $R^2$ and adjusted $R^2$ of 0.890 and 0.813, respectively. The *MAPE* was 13.8%. Speed of brushes, peeling time, and thawing incubation time significantly ($p < 0.05$) influenced the peeled surface area of cassava tubers. Higher *p*-values for the thawing temperature and some interaction terms suggested little impact on the peeled surface area of cassava tubers.

Figure 2 presents the surfaces plots for the peeled surface area as a function of the rotational speed of the brushes, peeling time, thawing temperature, and incubation time. The model was further verified with the normal probability plot for the externally studentized residuals. It was determined that most of the residuals were on a straight line (Figure 2d). This indicates the normal distribution of data. Furthermore, the plot of residuals versus predicted values, as presented in Figure 2e, shows no clear pattern among the data, which suggests the absence of biases.

**Table 3.** Experimental layout designed by Design Expert and its corresponding experimental and predicted values of responses.

| Run | Factor Variables | | | | Responses | | | |
|-----|---|---|---|---|---|---|---|---|
| | Rotational Speed of Brushes (rpm) | Peeling Time (min) | Thawing Temperature (°C) | Incubation Time (s) | Peeled Surface Area (%) | | Peel Loss (%) | |
| | $X_1$ | $X_2$ | $X_3$ | $X_4$ | Exp [a] | Pred [b] | Exp | Pred |
| 1 | 1000 | 4 | 60 | 30 | 94.9 | 84.7 | 30.4 | 31.3 |
| 2 | 700 | 4 | 60 | 30 | 48.0 | 40.2 | 9.2 | 8.7 |
| 3 | 1000 | 2 | 60 | 90 | 76.6 | 70.5 | 12.5 | 13.4 |
| 4 | 850 | 5 | 70 | 60 | 78.2 | 81.3 | 27.2 | 26.8 |
| 5 | 1000 | 2 | 80 | 90 | 72.2 | 89.2 | 34.8 | 34.8 |
| 6 | 700 | 4 | 80 | 30 | 26.8 | 28.2 | 5.6 | 6.0 |
| 7 | 550 | 3 | 70 | 60 | 24.4 | 30.3 | 6.8 | 7.6 |
| 8 | 700 | 2 | 80 | 90 | 50.9 | 54.0 | 6.2 | 6.5 |
| 9 | 850 | 3 | 70 | 60 | 63.5 | 78.9 | 26.0 | 24.5 |
| 10 | 700 | 4 | 80 | 90 | 81.1 | 76.8 | 37.0 | 36.4 |
| 11 | 700 | 2 | 60 | 30 | 14.9 | 24.3 | 4.3 | 4.7 |
| 12 | 1000 | 4 | 60 | 90 | 97.9 | 104.9 [c] | 23.7 | 23.7 |
| 13 | 1150 | 3 | 70 | 60 | 87.1 | 76.6 | 21.8 | 20.3 |
| 14 | 850 | 3 | 70 | 60 | 84.9 | 78.9 | 23.5 | 24.5 |
| 15 | 1000 | 2 | 60 | 30 | 22.8 | 21.2 | 5.3 | 5.4 |
| 16 | 700 | 4 | 60 | 90 | 87.2 | 88.8 | 30.4 | 30.7 |
| 17 | 850 | 3 | 70 | 60 | 92.3 | 78.9 | 24.1 | 24.5 |
| 18 | 850 | 3 | 90 | 60 | 89.3 | 82.3 | 17.8 | 17.5 |
| 19 | 1000 | 2 | 80 | 30 | 31.1 | 39.9 | 6.6 | 7.5 |
| 20 | 1000 | 4 | 80 | 30 | 60.8 | 72.8 | 9.2 | 9.3 |
| 21 | 850 | 3 | 70 | 120 | 92.8 | 86.8 | 31.9 | 31.6 |
| 22 | 850 | 3 | 70 | 0 | 20.7 | 22.2 | 4.2 | 3.7 |
| 23 | 700 | 2 | 60 | 90 | 35.5 | 35.3 | 10.8 | 10.3 |
| 24 | 850 | 3 | 70 | 60 | 63.5 | 78.9 | 26.0 | 24.5 |
| 25 | 1000 | 4 | 80 | 90 | 94.5 | 93.0 | 37.5 | 38.3 |
| 26 | 850 | 1 | 70 | 60 | 36.0 | 28.3 | 10.8 | 10.4 |
| 27 | 700 | 2 | 80 | 30 | 53.5 | 43.1 | 10.2 | 9.7 |
| 28 | 850 | 3 | 50 | 60 | 65.5 | 75.5 | 20.6 | 20.3 |
| 29 | 850 | 3 | 70 | 60 | 84.7 | 78.9 | 23.5 | 24.5 |
| 30 | 850 | 3 | 70 | 60 | 91.9 | 78.9 | 24.1 | 24.5 |

[a] Experimental value, [b] predicted value, and [c] error of mathematical equation resulted in value higher than 100. The mean absolute percentage error (*MAPE*) of peeled surface area and peel loss was 4.2 and 13.8%, respectively.

**Table 4.** ANOVA for reduced cubic equation for the effect of freeze–thaw pre-treatment (FTP), rotational speed of brushes, and peeling time on the peeled surface area (PSA) of cassava tubers.

| Source | Sum of Squares | Degree of Freedom | Mean Square | F Value | p Value |
|--------|---|---|---|---|---|
| Intercept | 18615.46 | 12.00 | 1551.29 | 11.51 | <0.0001 |
| $X_1$-Rotational speed of brushes | 3225.20 | 1.00 | 3225.20 | 23.93 | 0.0001 |
| $X_2$-Peeling time | 4214.43 | 1.00 | 4214.43 | 31.27 | <0.0001 |
| $X_3$-Thawing temperature | 68.81 | 1.00 | 68.81 | 0.51 | 0.4846 |
| $X_4$-Incubation time | 6252.99 | 1.00 | 6252.99 | 46.39 | <0.0001 |
| $X_1 X_2$ | 205.05 | 1.00 | 205.05 | 1.52 | 0.2342 |
| $X_1 X_4$ | 25.05 | 1.00 | 25.05 | 0.19 | 0.6718 |
| $X_2 X_3$ | 940.43 | 1.00 | 940.43 | 6.98 | 0.0171 |
| $X_2 X_4$ | 18.39 | 1.00 | 18.39 | 0.14 | 0.7164 |
| $X_1^2$ | 1131.89 | 1.00 | 1131.89 | 8.40 | 0.01 |
| $X_2^2$ | 1012.01 | 1.00 | 1012.01 | 7.51 | 0.014 |
| $X_4^2$ | 1041.20 | 1.00 | 1041.20 | 7.72 | 0.0129 |
| $X_1 X_2 X_4$ | 1116.43 | 1.00 | 1116.43 | 8.28 | 0.0104 |
| Residual | 2291.34 | 17.00 | 134.78 | – | – |
| Lack-of-fit | 1407.20 | 12.00 | 117.27 | 0.66 | 0.7412 |
| Pure error | 884.14 | 5.00 | 176.83 | – | – |
| Correction total | 20906.81 | 29.00 | – | – | – |

$R^2$, 0.890; Adjusted $R^2$, 0.813; and $p < 0.05$ indicates significance at the 95% level.

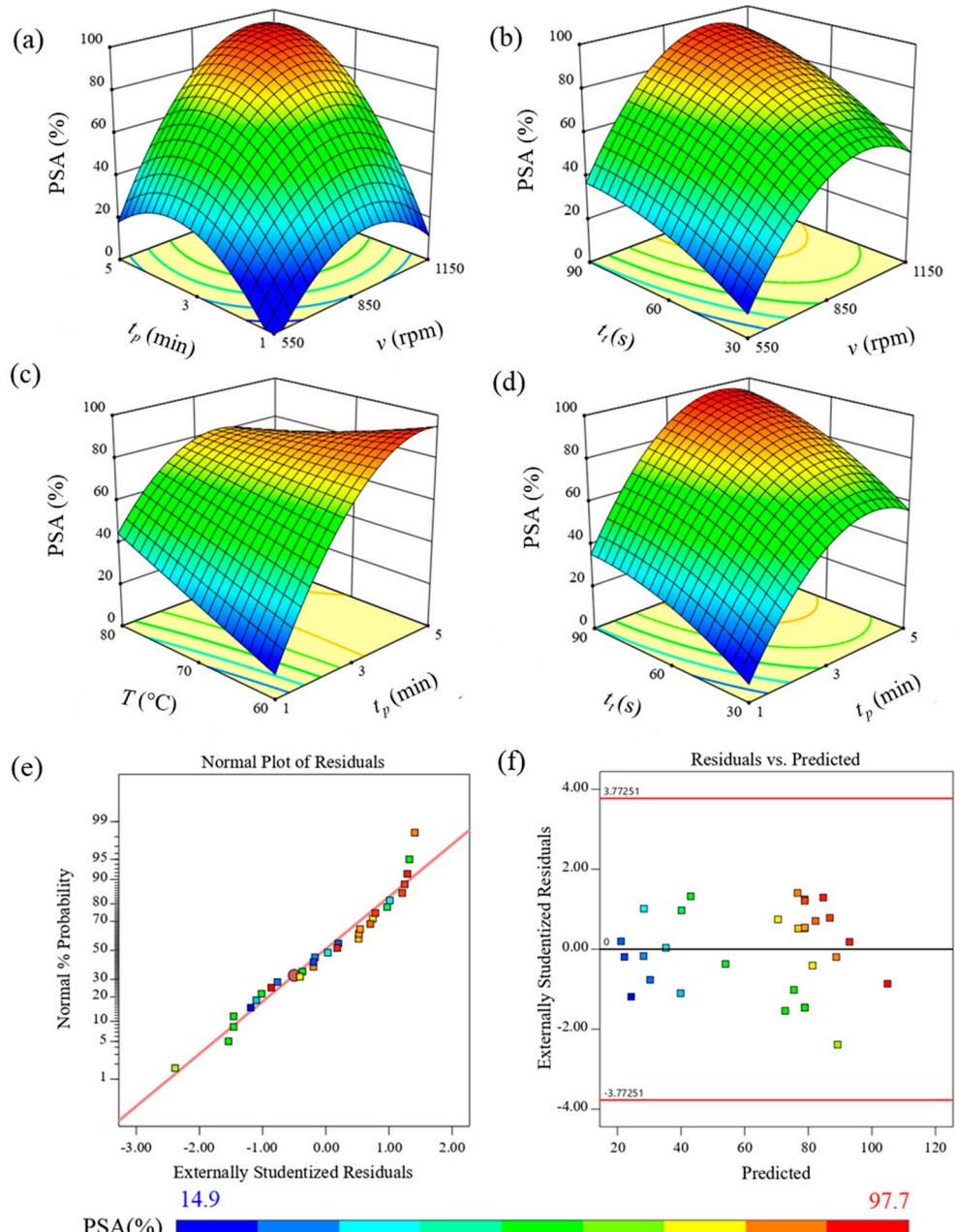

**Figure 2.** (**a**–**d**) Surface plots indicating the effect of the rotational speed of the brushes $v$ (rpm), peeling time $t_p$ (min), thawing temperature $T$ (°C), and incubation time $t_t$ (s) on the peeled surface area PSA (%) of cassava tubers while keeping the other variables constant. (**e**) Normal probability plot of the residuals of PSA. (**f**) Plot of residuals versus predicted values of PSA.

The mathematical equation obtained from RSM for the peel loss (*PL*) of cassava tubers is presented in Equation (4).

$$
\begin{aligned}
PL = {} & -739.80949 + 1.50160 \cdot v - 17.99998 \cdot t_p + 9.49199 \cdot T + 3.30502 \cdot t_t + 0.018653 \cdot v \\
& \cdot t_p - 0.015025 \cdot v \cdot T - 0.002785 \cdot v \cdot t_t + 0.542352 \cdot t_p \cdot T + 0.326473 \cdot t_p \\
& \cdot t_t - 0.076579 \cdot T \cdot t_t - 0.000954 \cdot v^2 - 1.47943 \cdot t_p^2 - 0.014144 \cdot T^2 \\
& - 0.001909 \cdot t_t^2 - 0.001361 \cdot v \cdot t_p \cdot T - 0.000889 \cdot v \cdot t_p \cdot t_t + 0.000078 \cdot v \quad \cdot T \cdot t_t \\
& + 0.007211 \cdot t_p \cdot T \cdot t_t + 0.000074 \cdot v^2 \cdot t_p + 0.0000088 \cdot v^2 \cdot T
\end{aligned}
\tag{4}
$$

where *PL* is the peel loss after FTP and the peeling process (%), $v$ is the rotational speed of the brushes (rpm), $t_p$ is the peeling time (min), $T$ is the thawing temperature (°C), and $t_t$ is the incubation time (s).

The results showed that increasing the rotational speed of the brushes, the peeling time and incubation time positively affected the peel loss. On the other hand, an increase in thawing temperature had a negative impact on the peel loss.

Table 5 presents the effects of the individual variables and their interaction on the peel loss. The accuracy of the model was indicated by $R^2$ and adjusted $R^2$ of 0.995 and 0.986, respectively. The *MAPE* was 4.20%. Rotational speed of the brushes, peeling time, and incubation time significantly ($p < 0.05$) influenced the peel loss of cassava tubers. Higher *p*-values for the thawing temperature and some interaction terms suggested little impact on the peel loss of cassava tuber.

**Table 5.** ANOVA for reduced cubic equation for the effect of FTP, the rotational speed of the brushes, and the peeling time on the peel loss (*PL*) of cassava tubers.

| Source | Sum of Squares | Degree of Freedom | Mean Square | F Value | p Value |
|---|---|---|---|---|---|
| Intercept | 3333.06 | 20.00 | 166.65 | 100.03 | <0.0001 |
| $X_1$-Rotational speed of brushes | 242.44 | 1.00 | 242.44 | 145.52 | <0.0001 |
| $X_2$-Peeling time | 134.14 | 1.00 | 134.14 | 80.51 | <0.0001 |
| $X_3$-Thawing temperature | 3.91 | 1.00 | 3.91 | 2.35 | 0.1597 |
| $X_4$-Incubation time | 1170.61 | 1.00 | 1170.61 | 702.65 | <0.0001 |
| $X_1X_2$ | 5.27 | 1.00 | 5.27 | 3.16 | 0.1090 |
| $X_1X_3$ | 8.98 | 1.00 | 8.98 | 5.39 | 0.0453 |
| $X_1X_4$ | 0.25 | 1.00 | 0.25 | 0.15 | 0.7062 |
| $X_2X_3$ | 52.94 | 1.00 | 52.94 | 31.78 | 0.0003 |
| $X_2X_4$ | 81.73 | 1.00 | 81.73 | 49.06 | <0.0001 |
| $X_3X_4$ | 193.85 | 1.00 | 193.85 | 116.35 | <0.0001 |
| $X_1{}^2$ | 193.16 | 1.00 | 193.16 | 115.94 | <0.0001 |
| $X_2{}^2$ | 60.03 | 1.00 | 60.03 | 36.03 | 0.0002 |
| $X_3{}^2$ | 54.87 | 1.00 | 54.87 | 32.94 | 0.0003 |
| $X_4{}^2$ | 80.95 | 1.00 | 80.95 | 48.59 | <0.0001 |
| $X_1X_2X_3$ | 66.69 | 1.00 | 66.69 | 40.03 | 0.0001 |
| $X_1X_2X_4$ | 256.22 | 1.00 | 256.22 | 153.80 | <0.0001 |
| $X_1X_3X_4$ | 198.61 | 1.00 | 198.61 | 119.21 | <0.0001 |
| $X_2X_3X_4$ | 74.87 | 1.00 | 74.87 | 44.94 | <0.0001 |
| $X_1{}^2X_2$ | 14.87 | 1.00 | 14.87 | 8.92 | 0.0153 |
| $X_1{}^2X_3$ | 20.77 | 1.00 | 20.77 | 12.47 | 0.0064 |
| Residual | 14.99 | 9.00 | 1.67 | – | – |
| Lack-of-fit | 8.20 | 4.00 | 2.05 | 1.51 | 0.3268 |
| Pure error | 6.79 | 5.00 | 1.36 | – | – |
| Correction total | 3348.05 | 29.00 | – | – | – |

$R^2$, 0.995; Adjusted $R^2$, 0.986; and $p < 0.05$ indicates significance at the 95% level.

Figure 3 shows the surfaces plots for peel loss as a function of the rotational speed of the brushes, peeling time, thawing temperature, and incubation time. The model was further analyzed with the normal probability plot for the externally studentized residuals. Similar to the peel loss, the data was normally distributed (Figure 3e) and there was no biases or clear patterns among the data (Figure 3f).

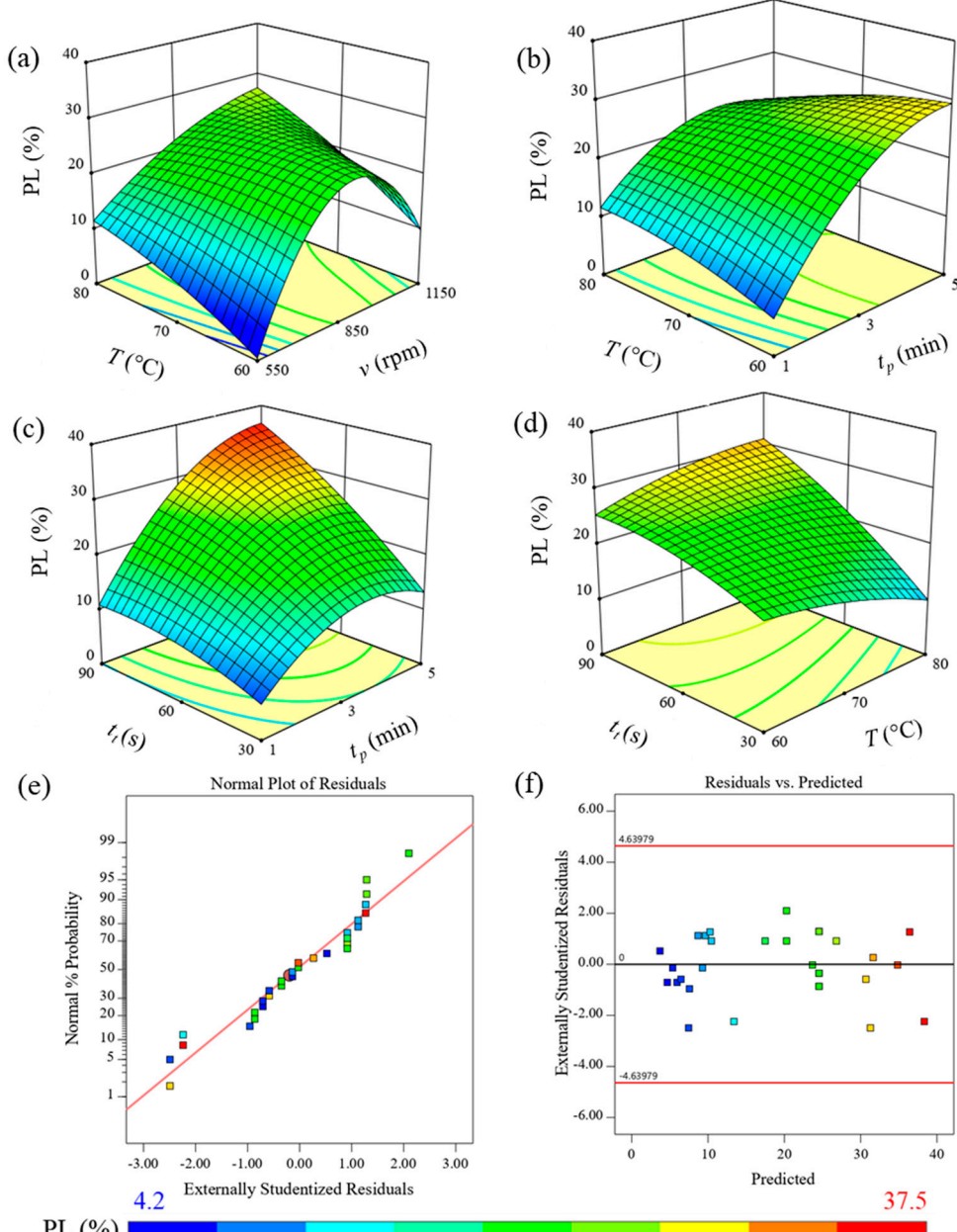

**Figure 3.** (**a–d**) Surface plots indicating the effect of the rotational speed of the brushes $v$ (rpm), peeling time $t_p$ (min), thawing temperature $T$ (°C), and incubation time $t_t$ (s) on the peel loss $PL$ (%) of cassava tubers, while keeping the other variables constant. (**e**) Normal probability plot of the residuals of PL. (**f**) Plot of residuals versus predicted values of PL.

Based on the model, an optimum peeling process was predicted for a peeled surface area of 99.5% and a peel loss of 19% at a rotational speed of 1000 rpm, peeling time of 3.4 min, thawing temperature of 59 °C, and incubation time of 90 s. The optimum peel loss of 19% of mechanical peeling was higher than the $PL_{man}$ of 14.5% obtained in manual peeling. The reason can be explained by variability in size and shape of cassava tubers. According to other studies, the proportion mass of peels in the tubers ranged from 15 to 20% [11,20]. Therefore, the peeled surface area of 99.5% and peel loss of 19% can be appropriate for the optimum peeling process in this study. Reaching 100% peeled surface area would increase the peel loss as well, which is not acceptable for cassava industries. To validate the prediction of the model, three trials were conducted at optimal conditions. The peeled surface area and the peel loss were 94.9 ± 2.6% and 21.7 ± 2.3%, respectively.

The rotational speed of the brushes and the peeling time were the most important variables for the peeling process. These findings are in line with other studies [20,24–27]. Based on Jimoh and Olukunle [26], cassava tuber loss increased and the peeling efficiency decreased by increasing the rotational speed of the rollers. Similarly, Ademosun et al. [24] stated that by increasing the rotational speed of the peeling chamber, the performance of the peeling machine was improved. Furthermore, Olukunle and Akinnuli [27] found that increasing the rotational speed of the brushes increased the peeling efficiency from 48.4 to 88.7%. In addition, by varying the speed from 140 to 160 rpm, the peeling efficiency increased from 81.2 to 91.4% [25].

The results indicate that FTP would improve the peeling of cassava tubers by softening and loosening the peels. The thermal shock induced by FTP resulted in a softening of the peels, and thus improved the peel separation. This finding is similar to results of Brown et al. [16] and Thomas et al. [17] where FTP has improved tomato peeling by loosening the peel.

There was no study to be found about the application of FTP on the peeling process of cassava tubers. Although the incubation time of the thawing treatment was short in this study, it is necessary to investigate the effect of FTP on the quality of cassava starch after the peeling process. The only thermal treatment applied to peel cassava tubers was a high-steam compression application [28]. It was determined that the time of steam exposure was one of the main problems of this method, because a long exposure time would cook the cassava tubers which affects their firmness, adhesiveness, and the springiness of the tubers [28].

### 3.3. Effect of FTP on the Quality of Cassava Tubers

According to Rahman [29], when food is frozen, the moisture in the food transforms into ice crystals that physically stress the food matrix. After thawing, the moisture is promptly separated from the food matrix (syneresis), which results in loosening and softening of the food texture and subsequently quality deterioration. Many researchers have been investigating the retrogradation and textural change of starch after FTP [30,31].

#### 3.3.1. Starch Content of Peeled Cassava Tubers

The starch content of manually peeled cassava tubers and tubers peeled after FTP are presented in Table 6. It was found that FTP did not significantly affect the starch content of the peeled tubers ($p < 0.05$). Similar results were also reported by Lee et al. [30], whereas no substantial difference in carbohydrate loss was detected in sweet potato after FTP.

**Table 6.** Comparison of starch content of peeled cassava tubers after FTP using the peeling machine at optimum conditions and manually peeled cassava tubers.

| Treatment | Total Starch Content (%, d.b.) |
|---|---|
| Manually peeled cassava tubers | 70.92 [a] ± 3.13 |
| Peeled cassava tubers after FTP using the peeling machine | 70.08 [a] ± 2.90 |

[a] Reported values are presented as mean values ± SD, n = 5. Mean values with the same letter in a column are not significantly different as indicated by T test ($p < 0.05$).

Freezing can influence textural and gelatinization characteristics of starch [30,31]. Moreover, it can also cause some alterations in the nutritional properties of the starch [32]. In addition, Oyeyinka et al. [33] stated that the freezing period could influence the functional, nutrient, and pasting properties of the product from the frozen cassava tubers. Therefore, it is necessary to further evaluate the quality of cassava tubers peeled by applying FTP in terms of resistance, damage, and gelatinized starch in future research.

### 3.3.2. Scanning Electron Microscopy of Peeled Cassava Tubers

According to other studies [34,35], freezing and thawing could form a solid matrix in foods due to syneresis. After the freezing and thawing treatment, ice cells were observed as cavities in the micrograph. The size of the cavity (ice crystal) in micrographs is positively related to the syneresis of starch [30]. In Figure 4, the scanning electron micrographs show a regular alignment of starch granules in both cassava tubers peeled manually and mechanically after FTP. It was determined that the FTP had no negative effect on the structure of the starch granules. No rupture and pores were observed in the starch granule structure due to FTP. In both micrographs, the starch granules had elliptic and oval shapes. The surface of the granules seemed to be smooth without any evidence of imperfections, fissures or pores at ×200 magnification (Figure 4b,d). However, at higher magnification (×2000) a slightly roughened surface could be observed in the starch granules after FTP (Figure 4a,c). Therefore, the structure of starch granules at higher magnification should be further investigated.

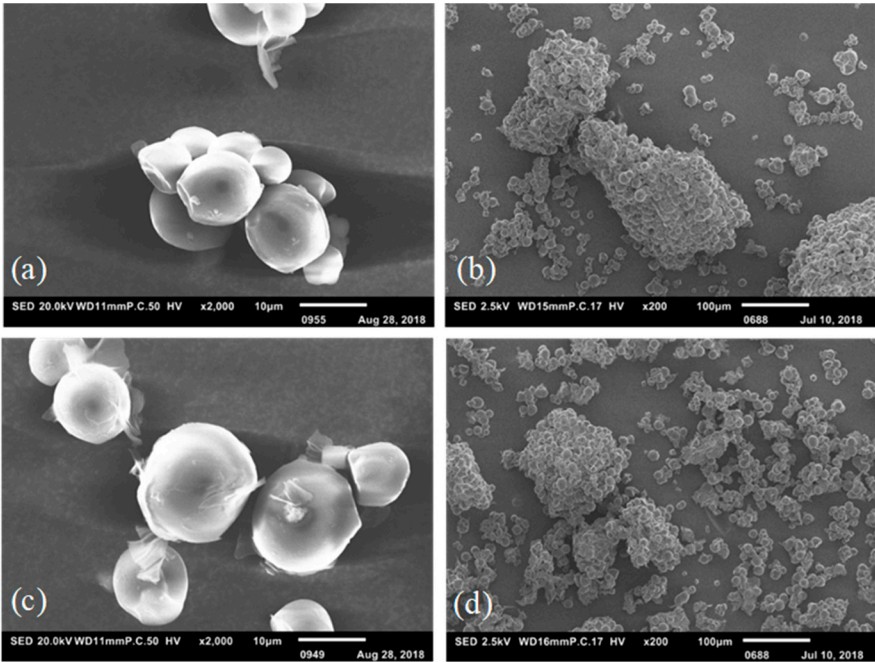

**Figure 4.** Scanning electron micrographs corresponding to peeled cassava tubers (**a**,**b**) with FTP using the peeling machine at optimum conditions and (**c**,**d**) manually peeled cassava tubers.

### 4. Conclusions

The application of a freeze–thaw pre-treatment for softening the peels proved to be an effective way to improve the mechanical peeling process of cassava tubers. The most efficient process conditions for an optimum peeling process were found to be a rotational brush speed of 1000 rpm, a peeling time of 3.4 min, thawing temperature of 59 °C, and an incubation time of 90 s, which resulted in a peeled surface area of 99.5% and a peel loss of 19%. The starch content and SEM study showed no negative effects of the freeze–thaw pre-treatment on the quality of cassava tubers.

For further studies, it is recommended to investigate the effects of different freezing duration, freezing temperature, and the use of liquid nitrogen in the freeze–thaw pre-treatment on the peeling process. The economic feasibility of the freeze–thaw pre-treatment for practical application should be studied because freezing operation can considerably increase the cost of processing.

**Author Contributions:** Conceptualization, Z.B., S.L. S.R., and J.M.; methodology, Z.B. and S.R.; validation, Z.B.; formal analysis, Z.B.; investigation, Z.B.; resources, J.M.; data curation, Z.B.; writing—original draft preparation, Z.B.; writing—review and editing, S.L, S.R., and J.M.; visualization, Z.B.; supervision, S.L. and J.M.; project administration, S.L. and J.M.; funding acquisition, S.L. and J.M.

**Funding:** This research was funded by German Federal Ministry of Education and Research (BMBF) under Project No. 031B0217 "CassavaUpgrade" and the Foundation fiat panis (Ulm, Germany) under Project No. 33/2015 "Evaluation and optimization of enzymatic cassava roots peeling".

**Acknowledgments:** The authors would like to thank the Institute of Food Processing Engineering and Powder Technology, University of Hohenheim, for the use of the scanning electron microscope, Jens Hartung for his suggestions for the improvement of the manuscript, and Sabine Nugent for English proofreading of this manuscript.

**Conflicts of Interest:** The authors declare no conflict of interest.

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
