# Peer review of "Freeze–Thaw Pre-Treatment of Cassava Tubers to Improve Efficiency of Mechanical Peeling"

_applsci, doi:10.3390/app9142856_

Round 1

Reviewer 1 Report

Abstract 

Line 10 - mention that the cassava peeling machine was an abrasive peeler. 

Line 10 - was peeling machine design and construction part of this study. I believe the peeling machine was already available. If a new machine was designed and constructed only for this experiment what was the basis of that? Best would have been using commercially available machines which are in use by industry.  

Introduction 

1. Line 52 - Objectives, please expand the objective section (maybe bullet points), to reflect all the sections of results (i.e. Section 3.3, 3.4, 3.5, has no mention in the objectives). It seems objective was also to understand the effect of FTP on functional quality and starch content of the. 

2. The freezing period can have a detrimental effect on processed products quality and quantity. Please also look at this paper: https://doi.org/10.1016/j.lwt.2018.10.004 , please state how long the freezing was done. 

Material and Methods 

Line 56 - Do we know the origin of the samples (country, variety, days after harvest etc)? If this info is available you can add it, otherwise, it is ok. 

Line 69 - what is PL(man) - please state clearly it is the mass of peel when carefully peel is removed. 

Line 70 - make the equation more visible and list variable of the equation clearly under as points, not in text form. Same for other equations in the paper. 

Figure 1: Please label the figure with its component. 

Line 87: For how long tubers stayed in the freezer, the length of the freezing period can have an n effect on the starch content and other properties. 

Line 94 - only 3 tubers ( Is this statistically sufficient ?) 

Please add a reference to section 2.5 and 2.6 in Objectives as mentioned previously. There is no mention in objectives about starch and SEM analysis. 

Line 123 - Dried sample for SEM, sample preparation is key for SEM analysis, how was sample dried (freeze dried ? or another supercritical drying), was the sample coated with a conductive metal (such as chrome or gold before SEM). - Please elaborate a little on the sample preparation for SEM. 

Line 129- Design-Expert rather than Expert Design. 

Result and Discussion 

Line 166-168 - Can you present this equation in a more readable manner. May reduce the decimal points to one or two. Same for the Eq. 4.

Conclusion

It will be premature to claim that (Line 291), that FTP is an effective way to improve mechanical peeling. I think not only future studies, but the practical limitation of freezing tonnes of cassava in large commercial operations should be highlighted. Such a freezing operation can significantly increase the cost of processing and does the increased cost can be levelled off in terms of saved losses. 

Author Response

We thank the reviewer for his/her comments and suggestions on our manuscript. We provide a response to each point in this attached document.

Reviewer 2 Report

The main objectives of this study were to investigate the impact of freeze-thawing on the peeling of cassava  tubers and to optimize the freeze-thaw pre-treatment to improve the peeling process. The operational parameters evaluated were the rotational speed of the brushes , peeling time , thawing temperature  and incubation time of the thawing treatment.Response surface methodology was applied to optimize FTP to improve the peeling process of 14 cassava tubers. Peeled surface area and peel loss were measured as the responses.

The paper is well done but I have some minor remark:

- there are some edit mistake

- the tab 1 is too large ( the  authors should editing the table in one page)

Author Response

(The authors gave the same response as above.)
